# The Design and Development of a Multi-HBV Antigen Encoded in Chimpanzee Adenoviral and Modified Vaccinia Ankara Viral Vectors; A Novel Therapeutic Vaccine Strategy against HBV

**DOI:** 10.3390/vaccines8020184

**Published:** 2020-04-14

**Authors:** Senthil K. Chinnakannan, Tamsin N. Cargill, Timothy A. Donnison, M. Azim Ansari, Sarah Sebastian, Lian Ni Lee, Claire Hutchings, Paul Klenerman, Mala K. Maini, Tom Evans, Eleanor Barnes

**Affiliations:** 1Peter Medawar Building, Nuffield Department of Medicine, University of Oxford, Oxford OX1 3SY, UK; senthil.chinnakannan@ndm.ox.ac.uk (S.K.C.); tamsin.cargill@bnc.ox.ac.uk (T.N.C.); timothy.donnison@ndm.ox.ac.uk (T.A.D.); lian.lee@ndm.ox.ac.uk (L.N.L.); claire.hutchings@ndm.ox.ac.uk (C.H.); paul.klenerman@medawar.ox.ac.uk (P.K.); 2Wellcome Centre for Human Genetics, University of Oxford, Oxford OX3 7BN, UK; ansari@well.ox.ac.uk; 3Vaccitech, The Oxford Science Park, The Schrodinger Building, Heatley Road, Oxford OX4 4GE, UK; Sarah.Sebastian@vaccitech.co.uk (S.S.); tom.evans@vaccitech.co.uk (T.E.); 4Division of Infection and Immunity, Institute of Immunity and Transplantation, University College London, London WC1E 6JF, UK; m.maini@ucl.ac.uk

**Keywords:** Hepatitis B virus (HBV), therapeutic HBV vaccine, T cell vaccine, chimpanzee adenovirus, ChAd, ChAdOx1, modified vaccinia Ankara, MVA

## Abstract

Chronic hepatitis B virus (HBV) infection affects 257 million people globally. Current therapies suppress HBV but viral rebound occurs on cessation of therapy; novel therapeutic strategies are urgently required. To develop a therapeutic HBV vaccine that can induce high magnitude T cells to all major HBV antigens, we have developed a novel HBV vaccine using chimpanzee adenovirus (ChAd) and modified vaccinia Ankara (MVA) viral vectors encoding multiple HBV antigens. ChAd vaccine alone generated very high magnitude HBV specific T cell responses to all HBV major antigens. The inclusion of a shark Invariant (SIi) chain genetic adjuvant significantly enhanced the magnitude of T-cells against HBV antigens. Compared to ChAd alone vaccination, ChAd-prime followed by MVA-boost vaccination further enhanced the magnitude and breadth of the vaccine induced T cell response. Intra-cellular cytokine staining study showed that HBV specific CD8+ and CD4+ T cells were polyfunctional, producing combinations of IFNγ, TNF-α, and IL-2. In summary, we have generated genetically adjuvanted ChAd and MVA vectored HBV vaccines with the potential to induce high-magnitude T cell responses through a prime-boost therapeutic vaccination approach. These pre-clinical studies pave the way for new studies of HBV therapeutic vaccination in humans with chronic hepatitis B infection.

## 1. Introduction

Chronic hepatitis B virus infection (CHBV) is a massive global health burden affecting 257 million people worldwide with an estimated 887,000 annual deaths due to liver cirrhosis and hepatocellular carcinoma (HCC) (WHO 2018). Currently available antiviral drugs suppress viral replication, but fail to suppress viral transcription and translation or eliminate covalently closed circular DNA (cccDNA), the viral DNA template. As a result, cessation of therapy results in viral rebound such that antiviral therapy is often continued indefinitely [1,2]. In addition, treatment does not eliminate the risk of developing HCC [3,4,5]. Therefore, there is an urgent need to develop effective therapeutic strategies to cure chronic HBV infection and to replace the current strategy of long-term antiviral treatment.

A robust T cell response is necessary for clearance of acute HBV. However, if infection persists, HBV specific T cells develop an exhausted phenotype and lose their full cytotoxic and proliferative functions [6]. Therapeutic vaccination, with the aim of inducing high-magnitude functional HBV T cells, is one strategy to induce immune control and functional cure of chronic HBV (defined by sustained HBsAg loss [7]).

We and others have previously shown that the viral vectors chimpanzee adenovirus (ChAd) and modified vaccinia Ankara (MVA) are the most effective clinical platform to generate high-magnitude, polyfunctional CD4+ and CD8+ T cell responses encoding HCV [8], malaria [9], and HIV [10] antigens. These vectors appear safe, and are both affordable and scalable. The use of chimpanzee adenoviral vectors overcomes the issue of pre-existing immunity to human adenoviral vectors that may otherwise limit vector efficacy in humans (reviewed in [11]). To generate a broad immune response targeting multiple HBV proteins, we designed a new HBV immunogen encompassing three full length HBV-antigens (precore/core, polymerase and preS1/preS2/surface) based on a patient genotype C sequence with maximum similarity to the genotype C consensus. The immunogen was linked to truncated shark class-II invariant chain (SIi) and tissue plasminogen activator (TPA) genetic sequences, and encoded into the chimpanzee adenovirus (ChAd) and modified vaccinia Ankara (MVA) viral vectors; these approaches have been shown to significantly enhance both CD8+ and CD4+ immune responses against encoded antigens [12,13]. We show that these vaccines can induce high magnitude polyfunctional T cells in immunocompetent uninfected mice against all major HBV immunogens, providing support for their clinical development in human trials.

## 2. Materials and Methods

### 2.1. HBV Immunogens

An HBV genotype C sequence (accession number: KJ173426 HBV isolate C2) isolated from an individual with HBV infection with maximum similarity to the consensus was used for designing HBV immunogens. First generation HBV immunogens (CP_mut_S, SIi-CP_mut_S and SIi-SCP_mut_) with and without class II shark invariant chain (SIi), were designed to encode precore (PreC), core, non-functional polymerase (P_mut_), and large surface antigen. These were then modified for incorporation into ChAdOx1/ChAdOx2 and modified vaccinia Ankara (MVA); SIi, PreC, core, P_mut_, NΔPreS1, and PreS2 were generated under a CMV (ChAd) or F11 (MVA) promoter to induce HBV specific T cells, whilst tissue plasminogen activator (TPA), CΔPreS1 and surface was generated separately using F2A (in ChAdOx1/ChAdOx2) or mH5 promoter (in MVA) in order to generate envelope protein and anti-HBV B cell responses. The order of the HBV core, pol and PreS1 and PreS2 immunogen sequence were rearranged within the ChAdOx1/ChAdOx2 and MVA to prevent the generation of immune responses to inter-gene regions when used in prime boost strategies.

### 2.2. Plasmids

Gene synthetic constructs, codon optimized for Homo sapiens (with GC content of 62% for pENTR4-SIi-CP_mut_S, 58% for pENTR4-SIi-CP_mut_TPA-S_(sh)_, 38% for pMVA-shuttle-SIi-CP_mut_TPA-S_(sh)_), encoding different HBV immunogens were generated using GeneArt, Thermo Fisher Scientific and cloned into either pENTR4 (downstream to the human cytomegalovirus long or short promoter and Kozak sequence) or the F11 locus based pMVA-shuttle vectors for generating ChAdOx1/ChAdOx2 and MVA recombinants, respectively. Plasmids for ChAdOx1/ChAdOx2 viral recombinants, encoding different HBV immunogens, were generated by moving the entire codon cassette from pENTR4 vector into the E1 locus of the replication deficient (lacking E1 and E3 region) ChAdOx1/ChAdOx2 genome using Thermo Fisher Scientific LR gateway cloning. Plasmids encoding the following HBV immunogens or HBV-polymerase were generated in the study (a) pENTR4-long CMV promoter encoding (i) SIi-CP_mut_S (ii) SIi-SCP_mut_ (iii) CP_mut_S (iii) SIi-CP_mut_TPA-S_(sh)_ (iv) FLAG-P_wt-2_ (v) FLAG-P_mut_ (vi) FLAG-P_mutΔ193–326_ (vii) FLAG-P_mutΔYMDD_ (b) pENTR4-short CMV promoter encoding SIi-CP_mut_TPA-S_(sh)_ (c) pChAdOx2 with long CMV promoter encoding (i) SIi-CP_mut_S (ii) CP_mut_S (iii) SIi-CP_mut_TPA-S_(sh)_ (d) pChAdOx2 with short CMV promoter encoding SIi-CP_mut_TPA-S_(sh)_ (e) pChAdOx1 with long CMV promoter encoding SIi-CP_mut_TPA-S_(sh)_ (f) pChAdOx1 with short CMV promoter encoding SIi-CP_mut_TPA-S_(sh)_. pCH-9/3091-P11 and pP_wt-1_ were plasmids used by Prof. Michael Nassal’s lab at University Hospital Freiburg, Germany for testing the functionality of different polymerase variants.

### 2.3. HBV Immunogen Expression Analysis in Western Blots

4 × 10^5^ HEK293A cells were plated on to 24 well plates. After overnight incubation, cells were transfected with 1μg of plasmids encoding SIi-CP_mut_S and SIi- SCP_mut_ using lipofectamine 2000 (Thermo Fisher Scientific, 11668019), according to manufacturer’s protocol. Twenty-four hours post-transfection, cells were lysed in 100 μL of SDS sample buffer (New England Biolabs, B7709S) and 1/3^rd^ of the protein samples were separated on 4%–15% SDS-PAGE gels (Biorad, 4561083) and sequentially immunoblotted with mouse anti-HepB pol (Santacruz, sc-81590), mouse anti-PreS1 (Santacruz, sc-57761), mouse anti-GAPDH (Novus biologicals, NB600-502) and sheep anti-mouse HRP (Amersham, NA931-100UL) antibodies.

### 2.4. Vaccines

All ChAdOx1/ChAdOx2 and MVA viral vectored vaccines were generated at the Viral Vector Core Facility (VVCF) of the Jenner Institute, University of Oxford, UK. ChAdOx1/ChAdOx2 vaccines were produced using the T-Rex-293 cell line (Thermo Fisher Scientific, Paisely, UK) and purified by cesium chloride centrifugation. MVA vaccines were produced using the DF-1 cell line and purified using sucrose cushion centrifugation.

### 2.5. Mutant HBV-Polymerase Functionality Analysis

Plasmid constructs encoding either wild-type (P_wt-1_/FLAG-P_wt-2_) or mutant (FLAG-P_mut_, FLAG-P_mutΔ193–326_, FLAG-P_mutΔYMDD_) HBV polymerase were co-transfected with pCH-9/3091-P11 into Huh7 human hepatoma cells. Four days post-transfection, cells were lysed with NP40 lysis buffer. Capsids from cytoplasmic lysates were separated by native agarose gel electrophoresis (NAGE) and blotted by capillary transfer in parallel onto PVDF and nylon membranes. PVDF blots were used to detect core protein in capsids by peroxidase-mediated chemiluminescent anti-HBc immunostaining (mAb 312-PO). Nylon blots were used to detect encapsidated HBV DNA by hybridization with a ^32^P-labeled HBV specific probe, after alkali-induced denaturation of the blotted capsids. Expression of the FLAG-tagged polymerase variants (P_wt-2_, P_mut_, P_mutΔ193–326_, and P_mutΔYMDD_) was verified by anti-FLAG (Sigma, F1804) Western blotting. 

### 2.6. Animal Experiments

Animal experiments were performed at the Biomedical Services Building, Oxford according to the UK Home Office Regulations (license numbers 30/2744 and 30/3293) and approved by the local ethical review board at the University of Oxford. All animal experiments were carried out in accordance with the UK Animals (Scientific Procedure) Act, 1986. 

#### 2.6.1. Mice

Outbred adult (age over 8 weeks old) female CD1 and inbred female adult BALBc and C57BL/6J mice were purchased from Charles River, UK. Male and female adult HHD mice [14] were a kind donation from Uzi Gileadi, Weatherall Institute of Molecular Medicine, University of Oxford. Mice were housed in individually ventilated cages in a containment level 2 facility.

#### 2.6.2. Intervention

Mice were vaccinated via intramuscular injections into the hind leg with chimpanzee-adenovirus (at a dose of 5 × 10^5^ or 4 × 10^7^ or 5 × 10^7^ infectious units (iu) in 50 μL per injection) and/or MVA (at a dose of 2 × 10^6^ plaque forming units (pfu) in 50 μL per injection) and sacrificed 7–14 days after last immunization.

#### 2.6.3. Experimental Design

Experiments using CD1, BALBc and C57BL/6J mice compared the results of two different vaccines or vaccination regimes. The sample size chosen for outbred CD1 mice (*n* = 10 per group) was higher than or inbred BALBc or C57BL/6J mice (*n* = 5 per group) based on assumed higher genetic variability in the former. All experiments included a group of unvaccinated animals as negative controls (*n* = 1–3 per experiment). Individual mice were not randomized to intervention and investigators were not blinded to the intervention group. The primary outcome measurement was the total magnitude of splenocyte responses as measured by IFNγ ELISpot assays.

For experiments using HHD mice [14] mice were randomized to intervention group using a computer-generated algorithm using the NC3Rs Experimental Design Assistant (EDA, https://www.nc3rs.org.uk/experimental-design-assistant-eda [15,16] and investigators were blinded to allocation during the study until data analysis had been completed. Sample size was calculated based on the total magnitude of splenocyte responses as measured by IFNγ ELISpot assays in experimental data from C57BL6/J mice.

### 2.7. Peptides

15-mer synthetic peptides, overlapping by 11 amino acids, spanning the entire HBV-immunogen (SIi-CP_mut_S) were obtained from Mimotopes, Australia. The peptides were then dissolved in DMSO and pooled as per requirements and stored at −80 °C. Before use, each pool was diluted in RPMI-1640 growth media at a concentration of 6 μg/mL.

### 2.8. Splenocyte and Intra Hepatic Lymphocyte Isolation

Spleen and perfused liver were collected from mice in phosphate buffered saline (PBS). Splenic lymphocytes were isolated by gentle mechanical disruption through a 40 μm cell strainer (Argos technologies) followed by one-minute exposure to ACK lysis buffer (Life Technologies). Intrahepatic lymphocytes were isolated by mechanical dislocation and Percoll gradient (GE healthcare) centrifuging followed by ACK lysis.

### 2.9. Ex-Vivo IFN ELISpots

2 × 10^5^ splenocytes or 1 × 10^5^ intrahepatic lymphocytes (IHLs) were plated on to ELISPot plates, pre-coated by overnight incubation at 4 °C with mouse anti-IFNγ monoclonal antibody (AN-18, MabTech), along with DMSO (1%) or non-HBV peptide pool (SIi, F2A and linker 1 and 2, at a concentration of 3 μg/mL) or HBV peptide pool (Core, Pol-1, Pol-2, Pol-3, Pol-4, PreS1/S2, and Surface at a concentration of 3 μg/mL) or a positive control mitogen (PHA or Concanavalin A at a concentration of 10 μg/mL and 12.5 μg/mL respectively). After a 16-h incubation at 37 °C, the plates were washed 7x with PBS and incubated with biotinylated mouse anti-IFNγ (R4-6A2, MabTech) for 2 h at room temperature, followed by 4x wash with PBS and alkaline phosphatase conjugated goat anti-biotin and incubation for 2 h at room temperature. The plates were then washed 4x with PBS and developed with NBT/BCIP substrate (34042, Thermo Fisher Scientific) until spots appeared on the wells. After a final wash with water and drying the spot forming units (SFU) per million cells from individual wells were counted on an automated ELISpot plate reader.

### 2.10. Intracellular Cytokine Staining

1 × 10^6^ splenocytes were stimulated with non-HBV peptide or HBV peptide pools, at a concentration of 2 μg/mL, for 5 h. GolgiPlug (BD Bioscience) was added 1 h after peptide stimulation. Cells were then surface stained for CD8 (eBioscience: PerCp-Cy5.5-anti-mouse CD8a, clone 53–6.7) and CD4 (eBioscience: AF-700-anti-mouse CD8a, clone GK1.5), fixed and permeabilized (fixation and permeabilization kit, BD Biosciences) and stained for IFN (eBioscience: PE-anti-mouse IFNγ, clone XMG1.2), TNF-α (eBioscience: FITC-anti-mouse TNF-α, clone MP6-XT22), and IL-2 (Biolegend: APC-anti-mouse IL-2, clone RMK-45) intracellular cytokines. Cells were acquired on a BD LSRII and analysed by FlowJo (Tree Star), Pestle and SPICEEv6 program.

### 2.11. ELISA

#### 2.11.1. Anti-HBs ELISA

ELISA plates were coated with recombinant HBV surface antigen (PIP002, Biorad) by overnight incubation at 4 °C. On the next day, plates were emptied and blocked with 10% skim milk powder diluted in 0.05% PBST (PBST with 0.05% Tween 20) for 1 h at 37 °C. After 3x wash with PBST, the plates were incubated with mouse sera diluted 1:3 in PBST. After 1-h incubation at 37 °C and 3x wash with PBST, plates were subsequently incubated with anti-mouse HRP antibody diluted 1:5000 in PBST. After final 3x wash with PBST, plates were incubated at room temperature with 1-step ultra-TMB-ELISA substrate (34029, Thermo Fisher Scientific) until colour development followed by addition of 1M sulphuric acid stop solution and measurement of ELISA optical density (OD) measurement at an absorbance of 450 nm.

#### 2.11.2. PreS1 and HBs Antigen Capture ELISA

PreS1 and HBs antigen capture ELISA was performed with following modifications to the standard protocol. Mouse monoclonal antibodies to PreS1 (Santacruz, sc-57761) and surface (GeneTex, GTX40707) were used as capture antibodies for PreS1 and HBs antigen capture, respectively. After blocking and wash step, cell culture supernatants were added to capture the PreS1 and HBs antigen, followed by wash and detection of captured antigens via HPR conjugated rabbit anti-HBs polyclonal antibody (GeneTex, GTX17448).

### 2.12. Statistical Analyses

Mann–Whitney-U tests or Kruskal–Wallis test with Dunn’s Multiple comparisons test were carried out using GraphPad PRISM version 7 program (GraphPad, La Jolla, CA).

## 3. Results

### 3.1. Generation of the HBV Immunogen

Strong T cell responses specific for epitopes from a variety of HBV proteins have been shown to play a major role in viral clearance of resolving acute HBV infection [17,18,19]. To generate a vaccine that induces multi-antigen specific T cell responses, we designed an HBV immunogen encompassing the major HBV proteins (precore (PreC), core, polymerase (P_mut_, with 8 point-mutations), preS1, preS2, and surface proteins), excluding the X-protein that has been associated with the development of HCC [20]. To design the HBV immunogen, we focused on HBV genotype C, highly prevalent in Asia. We downloaded 1447 HBV genotype C whole genome sequences from the Hepatitis B Virus database (HBVdb) [21,22], and aligned them using the MAFFT multiple sequence alignment program [23,24]. We then generated an HBV genotype C consensus sequence from this alignment and calculated the pairwise distance between the consensus sequence and all the sequences in the alignment. We selected an amino acid sequence isolated from an HBV infected individual closest to the consensus sequence (Figure 1a,b, accession number: KJ173426 HBV isolate C2). Alignment of precore, core, polymerase and large-surface protein sequences of the KJ173426 HBV isolate to the consensus protein sequences showed only one amino acid difference at position 321 in polymerase protein (leucine (L) in HBV genotype C consensus and phenylalanine (F) in KJ173426 HBV isolate C2)) (Figure 1c). This approach enabled us to generate an HBV immunogen, based on a real circulating HBV, which best represents all circulating HBV genotype C sequences.

### 3.2. Furin 2A Inclusion Enables the Generation of Two Separate Polypeptides from a Single Immunogen

To enable generation of a separate surface protein and the induction of anti-HBs antibodies, a furin 2A (F2A) ribosomal skipping peptide sequence was also included in the immunogen design (Figure 2a). Preliminary studies using multiple antigens that use a F2A peptide cleavage strategy have shown that the proximity of the antigen to the CMV promoter and F2A may influence the expression of the antigens (Prof. Sarah Gilbert personal communication). Therefore, two HBV immunogen cassettes that differed in the proximity of fused PreC/Core/P_mut_ protein (CP_mut_) and the large surface protein (S) to a long CMV promoter, SIi and F2A sequence were designed (Figure 2a) and the level of expression of CP_mut_ and S were tested in Western blotting experiments. Positioning the large surface protein codon next to the F2A cleavage site of SIi-CP_mut_S enabled high level of expression of separate CP_mut_ and S proteins (Figure 2b).

### 3.3. Addition of the Transmembrane Region of Shark Invariant Chain to the HBV Immunogen Enhances the Magnitude and Breadth of Vaccine Induced T Cell Responses

We generated an HBV immunogen with an amino-terminal tethered truncated shark Invariant chain (SIi) to function as a molecular enhancer encoded in ChAdOx2 [26], with a long CMV promoter. Tethering the 26 amino acids of the transmembrane region of SIi to the antigen has been shown to enhance vaccine-induced T cell responses [13]. We compared the capacity of ChAd vaccines with and without SIi, encoding the CP_mut_S-construct to induce HBV specific T cells (Figure 3a). Lymphocytes were extracted from the spleen and liver 14 days after intramuscular vaccination of outbred CD-1 mice and stimulated with pools of overlapping peptides corresponding to the vaccine immunogen. Lymphocytes from CD-1 mice vaccinated with SIi containing construct showed a significantly higher magnitude of total T cell responses to HBV proteins (median total response 2573 vs. 363 interferon-gamma (IFNγ) spot forming units (SFU) per million splenocytes, *p* = 0.001, and 3120 vs. 503 IFNγ SFU per million intrahepatic lymphocytes, *p* = 0.021, Figure 3b,c) and a higher number of positive peptide pools (median 4 vs 1 positive peptide pools in both splenocytes and liver lymphocytes, *p* = 0.002, Figure 3c). All further immunogens generated hereafter included SIi with the aim of enhancing the magnitude and breath of vaccine induced T cell responses.

### 3.4. HBV Immunogen Optimisation to Promote HBsAg Secretion

Next, we evaluated HBsAg protein secretion, that may be optimally required to generate hepatitis B surface antibodies (HbsAbs). In vitro analysis with the first-generation SIi-CP_mut_S immunogen encoding ChAd vaccine demonstrated that the SIi-CP_mut_S immunogen did not secrete HBs antigen (Figure 4a). To enable HBs secretion, we therefore re-engineered the large surface protein region of SIi-CP_mut_S, creating an artificial fusion protein S_(sh)_ that links the amino-terminal part of PreS1 with the small surface protein (S) via a small linker peptide sequence. The amino-terminus of S_(sh)_ was attached to the tissue plasminogen activator (TPA) leader sequence, a signal peptide sequence to facilitate secretion [27]. We relocated the PreS2 and the carboxyterminal part of PreS1 to the amino-terminus of F2A thereby retaining these antigens in the immunogen for targeting by T cells. We showed that both PreS1 and S were secreted in this second-generation construct and were readily detected in supernatants from ChAd HBV vaccine-infected HEK293A cells harvested 24 h post-infection and quantified in PreS1 and HBs antigen capture ELISA (Figure 4a).

### 3.5. Improving ChAd HBV Vaccine Stability

During vaccine manufacture we found that the immunogen (transgene) within ChAdOx2-SIi-CP_mut_S under a long CMV promotor was unstable, generating transgene deletion products during vector amplification in cell culture (data not shown). We therefore evaluated both an alternative vector (ChAdOx1) [28] and an alternative short CMV transgene promoter with the aim of improving transgene stability. We observed that ChAdOx1 encoding SIi-CP_mut_TPA-S_(sh)_ under a long CMV promoter, and both ChAdOx1 and ChAdOx2 encoding the SIi-CP_mut_TPA-S_(sh)_ transgene with a short CMV promoter (Figure 4b) were stable during multiple passages in cell culture (data not shown). 

We then compared immunogenicity of the stable vaccine constructs, ChAdOx1-LP-SIi-CP_mut_TPA-S_(sh)_, ChAdOx1-SP-SIi-CP_mut_TPA-S_(sh)_, and ChAdOx2-SP-SIi-CP_mut_TPA-S_(sh)_. To reduce the number of mice required per group in large comparative immunogenicity experiments, we used BALB/c, an inbred mouse strain, for this study. Inbred BALB/c mice intramuscularly vaccinated with ChAdOx1 short-CMV promoter-based SIi-CP_mut_TPA-S_(sh)_ vaccine induced a higher magnitude of HBV-specific T cells (median total responses of 4231 IFNγ SFU per million splenocytes) compared to the other two stable constructs (median total responses of 2980 and 2024 IFNγ SFU per million splenocytes, *** *p* < 0.001 for ChAdOx2 short-CMV promoter and ChAdOx1 long-CMV promoter, respectively) (Figure 4c,d). However, there were no differences in the breadth of HBV-specific T cell responses induced by these vaccines (median number of positive peptide pools 2–3 out of 7 possible pools, *p* = ns, Figure 4e). Based on stability and immunogenicity, ChAdOx1-SP-SIi-CP_mut_TPA-S_(sh)_ was chosen as the final version for further evaluation.

### 3.6. Mutant Polymerase (P_mut_) Encoded within the HBV-Immunogen is Non-Functional

To ensure that the polymerase in our immunogen was non-functional, eight amino acids (Figure 5a) previously shown to be essential for HBV-polymerase function were mutated in the polymerase sequence of SIi-CP_mut_S and SIi-CP_mut_TPA-S_(sh)_ immunogens [29,30,31,32,33]. To verify that these eight-point mutations were able to abolish the functionality of the vaccine-encoded polymerase, mammalian expression plasmids encoding mutant polymerase (P_mut_) and wild-type polymerase (P_wt_) were generated and tested in an assay that requires trans-complementation of a functional polymerase to rescue the replication of a polymerase-null HBV replicon. As expected, the wild-type polymerase was able to support replication of the polymerase-null HBV replicon. However, the mutant polymerase did not support replication (Figure 5b), indicating that the same mutant polymerase encoded in the HBV-vaccine was non-functional.

### 3.7. Vaccinating with MVA-SIi-CP_mut_TPA-S_(sh)_ 7–8 weeks after ChAdOx1-SP-SIi-CP_mut_TPA-S_(sh)_ Boosts the Magnitude of Vaccine-Induced T cell Responses

MVA vectored vaccines have been shown to boost the magnitude of adenoviral vaccine-induced T cell responses to several immunogens in mice [34,35,36] and humans [8,37,38,39,40,41]. We therefore designed an MVA encoding SIi-CP_mut_TPA-S_(sh)_ for use in a heterologous boost regimen following ChAdOx1-SP-SIi-CP_mut_TPA-S_(sh)_ prime vaccination. All the HBV genetic material that was included in the SIi-CP_mut_TPA-S_(sh)_ ChAd vaccine was retained in the MVA vectored vaccine, but the genes were rearranged so that “artificial” junctional non-HBV epitopes that may induce T cells following ChAd priming could not be enhanced by a heterologous MVA boost.

Inbred C57BL/6J mice were intramuscularly vaccinated with either ChAdOx1-SP-SIi-CP_mut_TPA-S_(sh)_ alone or ChAdOx1-SP-SIi-CP_mut_TPA-S_(sh)_ (Figure 6a) followed 7–8 weeks later by MVA-SIi-CP_mut_TPA-S_(sh)_ (Figure 6b). Splenocytes were harvested 14 days after the last vaccination and stimulated with pools of overlapping peptide corresponding to the vaccine immunogen in ELISpot assays. For both vaccine regimes, the dominant responses were seen to peptide pools corresponding to the first half of polymerase (Pol1 and Pol2) and surface (PreS1/PreS2 and Surface) (Figure 6c). There were no observable responses towards the non-HBV vaccine components (molecular enhancer and linker regions, Figure 6c). ChAdOx1-SP-SIi-CP_mut_TPA-S_(sh)_ prime vaccination alone induced lower-magnitude T cell responses than when ChAdOx1-SP-SIi-CP_mut_TPA-S_(sh)_ prime was followed by MVA-SIi-CP_mut_TPA-S_(sh)_ boost vaccination (median total response 438 vs 5325 IFNγ SFU per million splenocytes, *p* = 0.0079, Figure 6c,d). The breadth of vaccine-induced T cell responses was also increased by the addition of MVA-SIi-CP_mut_TPA-S_(sh)_ (median number of positive peptide pools 2 vs. 6 out of 7 possible pools, *p* = 0.0159, Figure 6e).

To further investigate the nature of vaccine-induced T cell responses, we detected cytokine production by intracellular staining after stimulation with pools of overlapping peptides corresponding to the vaccine insert sequence. In the CD8+ T cell compartment (Figure 6f), mice that had received ChAdOx1-SP-SIi-CP_mut_TPA-S_(sh)_ followed by MVA-SIi-CP_mut_TPA-S_(sh)_ compared to ChAdOx1-SP-SIi-CP_mut_TPA-S_(sh)_ alone had a higher percentage of CD8+ T cells that produced IFNγ to the polymerase 1 + 2 pool (1.96 vs. 7.1%, *p* = 0.0159) and the polymerase 3 + 4 pool (0.12 vs. 1.79%, *p* = 0.0317) and tumour necrosis factor-alpha (TNFα) to the polymerase 1 + 2 pool (0.72 vs. 1.34%, *p* = 0.159). With both vaccine regimens, nearly three-quarters or more of the vaccine induced CD8+ T cells were single cytokine producers (Figure 6g). In the CD4+ compartment (Figure 6h), mice that had received the ChAdOx1-SIi-CP_mut_TPA-S_(sh)_ followed by MVA-SIi-CP_mut_TPA-S_(sh)_ had a higher percentage of CD4+ T cells that produced either TNFα or interleukin-2 (IL-2) to the surface pool (0.018 vs. 0.139, *p* = 0.0317 and 0 vs. 0.11, *p* = 0.0476 respectively). With both vaccine regimens over half of the vaccine induced CD4+ T cells were single cytokine producers with the rest being double or triple producers of IFNγ, TNFα or IL-2, Figure 6i).

### 3.8. T cell Responses Towards HBV-Core Peptides Can Be Induced in HHD Mice

We sought to investigate whether HBV core-specific T cell responses could be detected in humanized mice, as they were undetectable in C57BL/6J mice. We vaccinated HLA-A2 restricted HHD mice [14] to determine whether T cell responses to HBV-core peptides could be induced through epitope presentation on human HLA-A2. Mice were vaccinated with ChAdOx1-SP-SIi-CP_mut_TPA-S_(sh)_ or ChAdOx1-GFP followed after 4 weeks by MVA-SIi-CP_mut_TPA-S_(sh)_ (Figure 7a). Splenocytes were harvested 7 days after the last vaccination and stimulated with pools of overlapping peptides corresponding to the HBV immunogen. T cell responses towards HBV-core, polymerase and surface overlapping peptides were detected (Figure 7b). Mice that received ChAdOx1-SIi-CP_mut_TPA-S_(sh)_ prime and MVA-SIi-CP_mut_TPA-S_(sh)_ had significantly higher-magnitude T cell responses (median total response 57.5 vs 517.5 IFNγ SFU per million splenocytes, *p* = 0.0007, Figure 7c) and a higher number of positive peptide pools (median 4 vs 1 positive peptide pools, *p* = 0.0005, Figure 7d) as compared to mice vaccinated with ChAdOx1-GFP prime and MVA-SIi-CP_mut_TPA-S_(sh)_ boost.

### 3.9. Vaccination with MVA-SIi-CP_mut_TPA-S_(sh)_ 7–8 Weeks after ChAdOx1-SP-SIi-CP_mut_TPA-S_(sh)_ Induces HBs-Antibody

Vaccine-induced anti-HBs antibody generation is desirable to target and neutralize any circulating HBV viral particles. To investigate whether anti-HBs antibody was present in the sera following vaccination with either ChAdOx1-SP-SIi-CP_mut_TPA-S_(sh)_ alone or ChAdOx1-SP-SIi-CP_mut_TPA-S_(sh)_ followed 7–8 weeks later by MVA-SIi-CP_mut_TPA-S_(sh)_ sera were obtained 2 weeks after the last vaccination in both groups and tested by ELISA. We did not detect anti-HBsAb in mice receiving high-dose ChAdOx1-SP-SIi-CP_mut_TPA-S_(sh)_ followed by MVA-SIi-CP_mut_TPA-S_(sh)_ or MVA-SIi-CP_mut_TPA-S_(sh)_ alone (Figure 8). However, anti-HBs levels were detected in mice vaccinated with low dose ChAdOx1-SP-SIi-CP_mut_TPA-S_(sh)_ followed by MVA-SIi-CP_mut_TPA-S_(sh)_ (median 86.05 ug/mL HBsAb).

## 4. Discussion

We describe the design and development of an HBV therapeutic vaccine strategy using ChAd and MVA viral vectors, encoding all major HBV antigens, using a heterologous prime boost strategy. We show that this strategy induces high-magnitude and broad HBV-specific T cell responses against all encoded antigens in immuno-competent mice.

Therapeutic vaccination has been tested previously in patients with chronic HBV with limited success (reviewed in [42]). One reason for this might be that the T cell responses, especially CD8+ T cells, induced by these vaccines were not potent enough to provide immunological control of HBV. Our vaccination strategy incorporates several features that augment antigen-specific CD4+ and CD8+ T cell responses, thereby increasing the chance of inducing functional cure in chronic HBV.

First, we use ChAd and MVA viral vectors in a heterologous prime boost regimen and show that the magnitude and breadth of CD8+ and CD4+ T cell responses primed by ChAdOx1-SP-SIi-CP_mut_TPA-S_(sh)_are increased in response to a boost vaccination with MVA-SIi-CP_mut_TPA-S_(sh)_. This is in keeping with the large body of data supporting the augmentation of T cell responses by MVA after a ChAd prime, in heterologous prime boost strategies in both mice and humans [8,9,10,34,35,36,43].

Second, we show that the addition of the truncated shark class-II invariant chain (SIi) to the immunogen enhances the magnitude of the vaccine-induced T cell response, as has been reported in a recent study of ChAd and MVA viral vectored malaria vaccines in mice [13].

Third, our vaccine immunogen covers the major HBV regions coding for the core, polymerase and surface proteins, where previous vaccinations were mostly restricted to conserved regions only or the surface protein alone (reviewed in [42]). We show that in HHD mice (which are transgenic for human HLA-A2.1), T cell responses to peptides from the core, polymerase and surface regions are detectable. Evidence suggests that multi-specific T cell responses are required for the successful immunological control of acute HBV infection [44]. Therefore, expanding the number of potential epitopes presented through vaccination may emulate the multi-specific T cell response seen in a resolving acute infection.

There are several potential limitations with our current ChAd and MVA vaccination strategy that will need further work to resolve. We detected core-specific T cell responses in HHD mice but not in C57BL/6J mice after vaccination. This may be due to competition with immunodominant epitopes in C57BL/6J mice or due to a separate mechanism. Further work in mice and immunogenicity studies in human trials will be required to understand this better.

Although we did detect anti-HBsAb in response to vaccination in some C57BL/6J mice, this was not ubiquitous, and titers varied. We show that the small S protein is secreted in vitro and that CD4+ surface-specific T cell responses are induced by vaccination. However, it is possible that CD4 immunodominant responses lie in regions other than the HBV surface region of the immunogen required for HBsAb generation and that antibodies are preferentially produced in response to the PreS1 region. Investigations of the surface-specific CD4+ T cell responses at the epitope level are needed.

Ultimately our HBV therapeutic vaccine strategy requires evaluation in animal models of persistent HBV infection and in people with chronic HBV to understand its potential to induce a functional cure. In persistent HBV infection, immune checkpoints are upregulated, and HBV specific T cells are functionally exhausted (reviewed in [6]). Our vaccine could be used in combination with other immune-modulating therapies such as checkpoint inhibitors to overcome this. Indeed, in the woodchuck hepatitis model, vaccination in combination with checkpoint modulator therapy showed promise in achieving a functional cure in chronically infected woodchucks [45].

The HBV immunogen encoded in our ChAd and MVA vaccines has been designed based on genotype C consensus, aiming to target regions with high genotype C prevalence such as South-East Asia, where the virus is endemic [46]. However, the cross-reactive potential and suitability of a genotype C immunogen-based vaccine in regions with other genotypes also requires further investigation.

In summary, we have described an HBV therapeutic vaccine strategy using ChAd and MVA viral vectors, that induces high-magnitude, broad, polyfunctional HBV specific T cell responses. This regimen is a promising candidate for therapeutic vaccination trials in people with chronic HBV.

## Figures and Tables

**Figure 1 vaccines-08-00184-f001:**
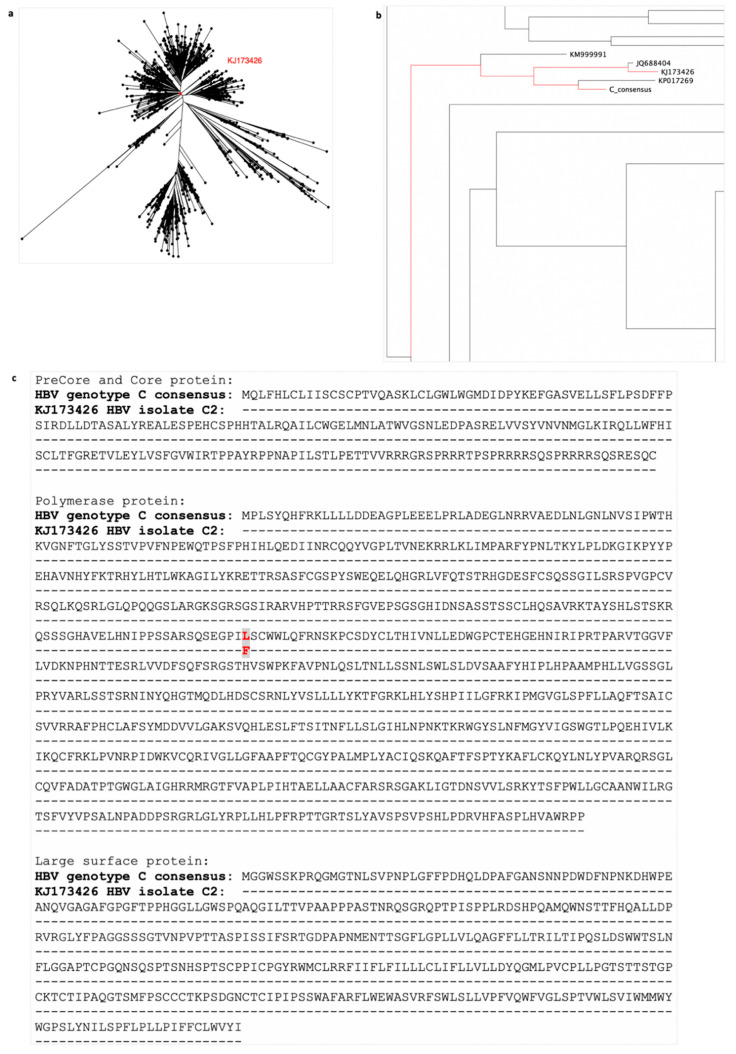
Selection of a patient’s hepatitis B virus (HBV) sequence for HBV genotype C immunogen design. 1447 HBV genotype C whole genome sequences downloaded from hepatitis B Virus database (HBVdb), were aligned using MAFFT and used to generate a consensus sequence for HBV genotype C (phylogenetic tree of all sequences shown in (**a**). The pairwise distance between each sequence in the alignment and the consensus sequence was calculated. The sequence isolated from a person infected with HBV with the closest amino acid sequence to the consensus was selected (accession number: KJ173426 HBV isolate C2). The phylogenetic tree shows the relationship of the chosen sequence (KJ173426 highlighted red) with all downloaded HBV genotype C sequences in (**a**) and to other closely related sequences (**b**). Sequence accession number KP017269 is not derived from an infected person (it is a consensus sequence) and so was disregarded. Alignment of precore/core/polymerase/large-surface protein of the consensus and selected KJ173426 sequence (**c**) showed one amino acid difference, at positions 321 in polymerase protein (leucine (L) in HBV genotype C consensus and phenylalanine (F) in KJ173426 HBV isolate C2), highlighted by grey box with red letters.

**Figure 2 vaccines-08-00184-f002:**
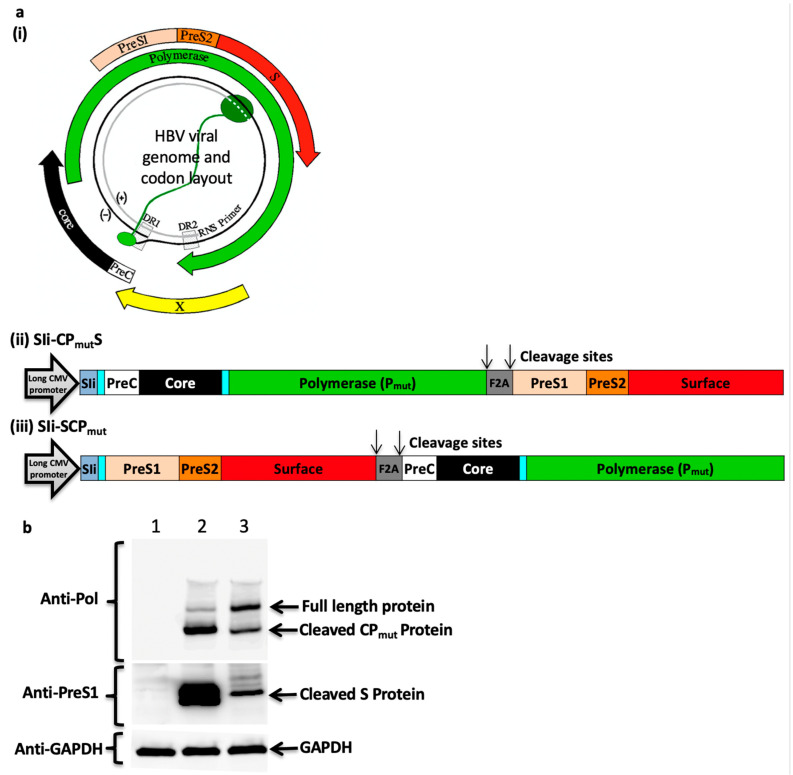
HBV immunogen design and in vitro protein expression analysis. (**a**) HBV viral genome and codon layout (**a, i**) (illustration from [25]). HBV genome comprise of a partially double stranded circular DNA, of approximately 3.2 kilobase (kb) pairs. It encodes 4 coding regions; precore along with core, polymerase, surface proteins (three forms of the surface proteins, L, M, and S, where the L-form is composed on PreS1, PreS2, and S, the M-form is composed of PreS2 and S) and x-protein. HBV immunogens SIi-CP_mut_S (**a, ii**) and SIi-SCP_mut_ (**a, iii**) were designed to encode HBV precore (PreC), core, polymerase (P_mut_), PreS1, PreS2, and surface proteins and non-HBV regions (comprising of a truncated shark Invariant chain (SIi), two linkers and a Furin 2A (F2A) peptide sequence). The preS1/preS2/surface region was positioned at carboxy-terminus of layout 1 and at the amino-terminus of layout 2. Within the mammalian expression cassette, the immunogen sequence was placed in between a long CMV promoter and BGH poly sequence. (**b**) Plasmids encoding SIi-CP_mut_S and SIi- SCP_mut_ were transfected into HEK293A cells. 24 h post-transfection, cells were lysed and the lysates were analysed in western blot experiments using mouse anti-HBV-PreS1 and mouse anti-HBV-Polymerase antibodies. Blots probed with mouse anti-GAPDH served as loading controls. Lane 1: cell lysate from un-transfected cells, lane 2 and lane 3: cell lysates from cells transfected with plasmids encoding SIi-CP_mut_S and SIi-SCP_mut_, respectively.

**Figure 3 vaccines-08-00184-f003:**
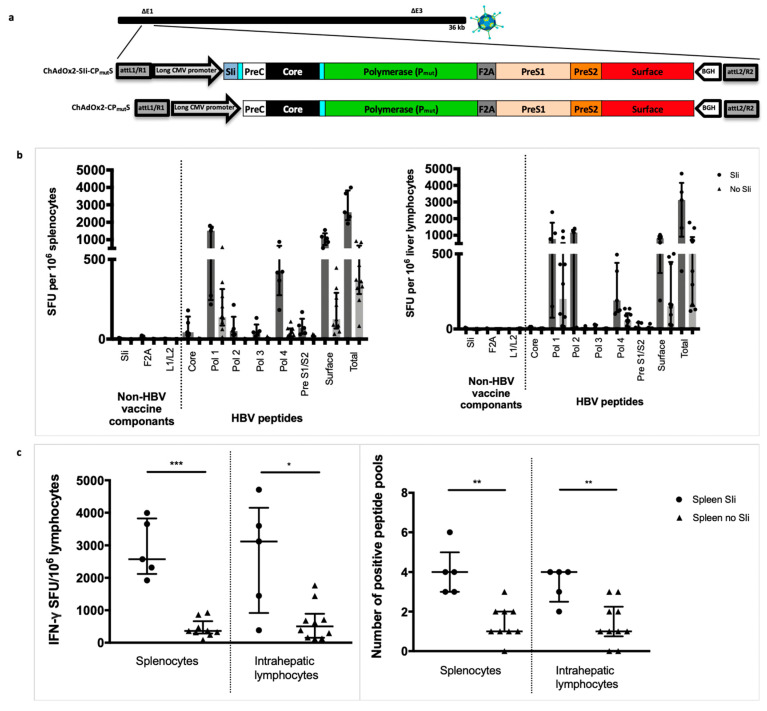
Tethering transmembrane region of shark invariant chain to the HBV immunogen enhances the magnitude and breadth of vaccine induced T cell responses. Long-CMV promoter (LP) based mammalian expression cassettes, encoding SIi-CP_mut_S or CP_mut_S immunogens, were inserted into E1 locus of replication deficient ChAdOx2 vector and recombinant ChAdOx2-SIi-CP_mut_S and ChAdOx2-CP_mut_S viruses were generated in T-REx™-293 cells (**a**). CD1 mice were vaccinated intramuscularly with 5 × 10^7^ infectious units of ChAdOx2 encoding either SIi-CP_mut_S (*n* = 5) or CP_mut_S (*n*= 9–10. Splenocytes (left panel) and intrahepatic lymphocytes (right panel) were extracted 14 days after vaccination and plated in interferon-gamma (IFNγ) ELISpot assays with pools of overlapping peptides corresponding to the vaccine immunogen (**b**). T cell response magnitude was measured by the number of IFNγ spot forming units (SFU) per million lymphocytes (**c**, left panel) and breath was measured by the number of positive peptide pools (**c**, right panel, over 100 IFNγ SFU/million cells defined as positive). Median and interquartile ranges are shown. Mann–Whitney tests were used for statistical comparison of medians between groups. * *p* < 0.05, ** *p* < 0.01, *** *p* < 0.001.

**Figure 4 vaccines-08-00184-f004:**
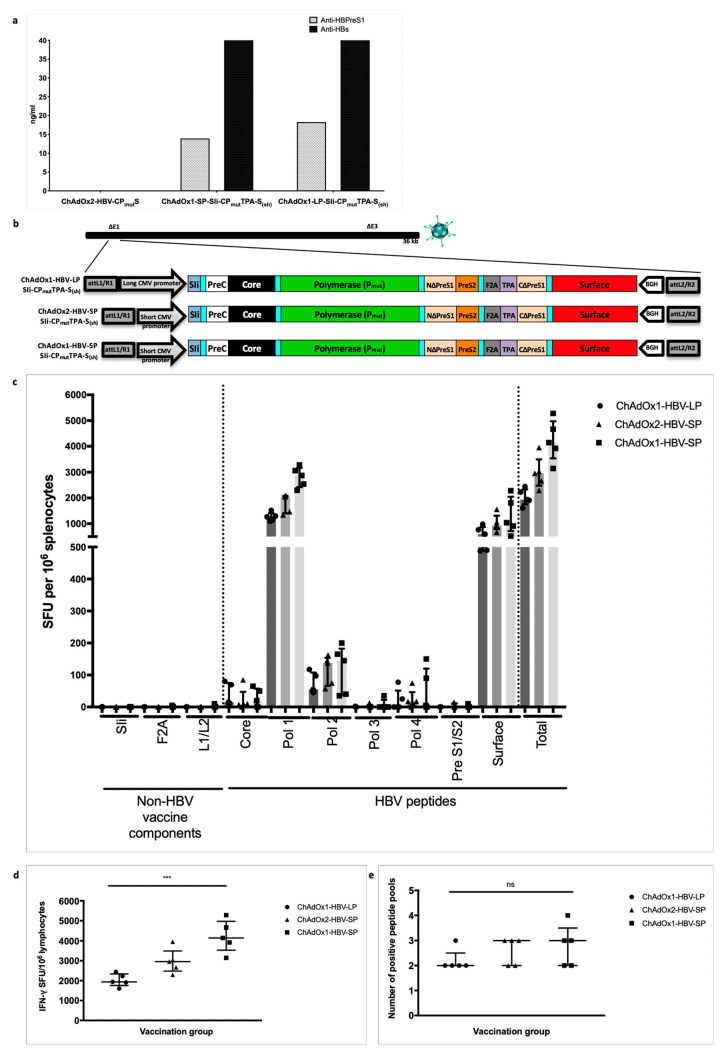
Second-generation HBV immunogen SIi-CP_mut_TPA-S_(sh)_ design and immunogenicity of chimpanzee adenovirus (ChAd) vaccine candidates in inbred Balb/c mice. Supernatants from chimp-adenoviral HBV vaccine infected HEK293A cells were harvested 24 h post-infection and quantified in PreS1 and HBs antigen capture ELISA. PreS1 and S antigen quantities based on commercial standard recombinant HBV surface antigen (PIP002, Biorad) obtained from different chimp-adenoviral HBV vaccines, above background value are shown (**a**). Long-CMV promoter (LP) or short-CMV promoter (SP) based mammalian expression cassettes, encoding SIi-CP_mut_TPA-S_(sh)_ immunogen were inserted into E1 locus of replication deficient ChAdOx1 or ChAdOx2 vector and four recombinant ChAdOx1-LP-SIi-CP_mut_TPA-S_(sh)_, ChAdOx1-SP-SIi-CP_mut_TPA-S_(sh)_, ChAdOx2-LP-SIi-CP_mut_TPA-S_(sh)_ and ChAdOx2-SP-SIi-CP_mut_TPA-S_(sh)_ viruses were generated in T-REx™-293 cells (**b**). Subsequently, all four recombinant viruses were screened by PCR, for stability analysis. Except ChAdOx2-LP-SIi-CP_mut_TPA-S_(sh)_, other three viruses showed positive PCR result for the presence of an intact HBV-immunogen cassette. BALB/c mice were vaccinated intramuscularly with 5 × 10^7^ infectious units of ChAdOx1-LP-SIi-CP_mut_TPA-S_(sh)_, or ChAdOx1-SP-SIi-CP_mut_TPA-S_(sh)_ or ChAdOx2-SP-SIi-CP_mut_TPA-S_(sh)_ (*n* = 5 per group). Splenocytes were extracted 14 days after vaccination and plated in IFNγ ELISpot assays with pools of overlapping peptides corresponding to the vaccine immunogen (**c**). T cell response magnitude was measured by the number of IFNγ SFU per million lymphocytes (**d**) and breath was measured by the number of positive peptide pools (**e**) over 100 IFNγ SFU/million cells defined as positive). Median and interquartile ranges are shown. Kruskal–Wallis tests were used for statistical comparison of the three vaccine groups. *** *p* < 0.001, ns = not significant.

**Figure 5 vaccines-08-00184-f005:**
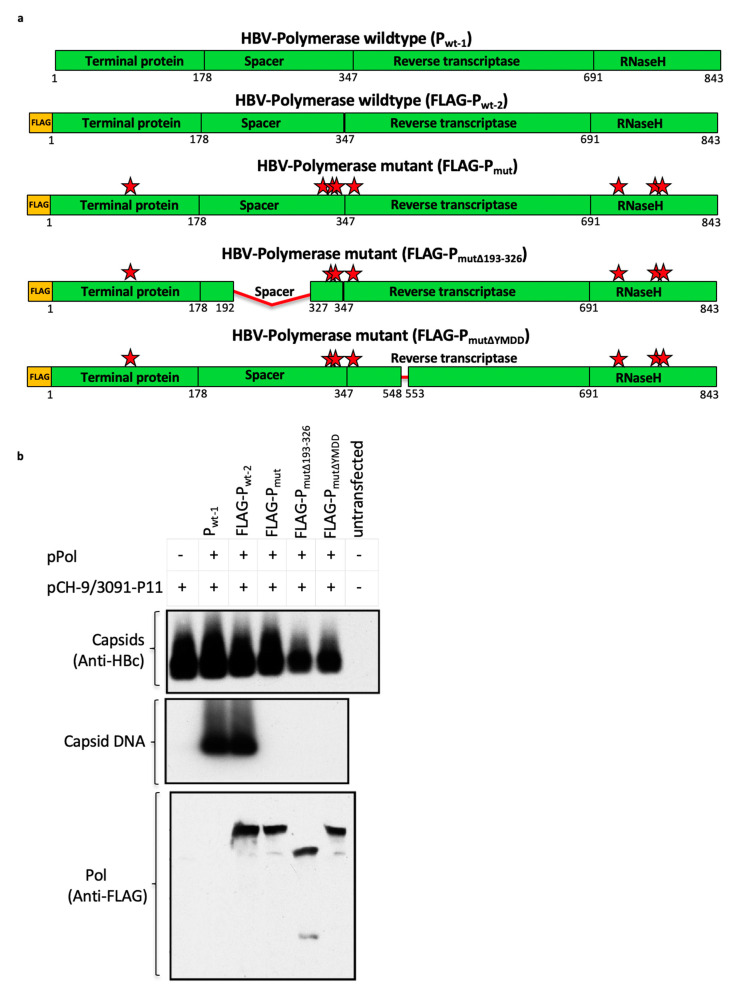
Mutant HBV polymerase (P_mut_) encoded by the chimpanzee-adenovirus and modified vaccinia Ankara (MVA) viral vectored HBV vaccines is non-functional. Schematic representation of wild-type (P_wt-1_/FLAG-P_wt-2_) and mutant (FLAG-P_mut_, FLAG-P_Δmut_, FLAG-P_mutΔYMDD_) HBV polymerases used to evaluate the possibility of abolishing the functionality of the HBV polymerase encoded in the vaccine using eight-point mutations (FLAG-P_mut_; alanine substitutions at positions Y63, C323, C334, C338, C352, R714, D777, R781, represented by red stars) on its own, or in combination with either 193–326 spacer region deletion (FLAG-P_mutΔ193–326_) or 549–552 YMDD motif deletion (FLAG-P_mutΔYMDD_) (**a**). Mammalian expression plasmids, each encoding a wild-type (P_wt-1_ and FLAG-P_wt-2_) or a mutant (FLAG-P_mut_, FLAG-P_mutΔ193–326_, FLAG-P_mutΔYMDD_) HBV polymerase, were generated and tested in a trans-complementation assay that requires co-transfection of a plasmid encoding functional HBV-polymerase, to rescue the replication of polymerase deficient HBV-genome (encoded via plasmid, pCH-9/3091-P11) (**b**). Plasmid constructs, as indicated in the top of the figure, were co-transfected into Huh7 cells. 4 days post-transfection cells were lysed, capsids were isolated and the levels of capsid protein and capsid DNA from each sample were analyzed in Western blot and Southern blot, using anti-capsid antibody and ^32^P-labelled HBV specific probe, respectively. Western blot probed with anti-FLAG antibody confirmed equivalent level of expression HBV polymerase in samples receiving FLAG-P_wt-2_, FLAG-P_mut_, FLAG-P_mutΔ193–326_, and FLAG-P_mutΔYMDD_.

**Figure 6 vaccines-08-00184-f006:**
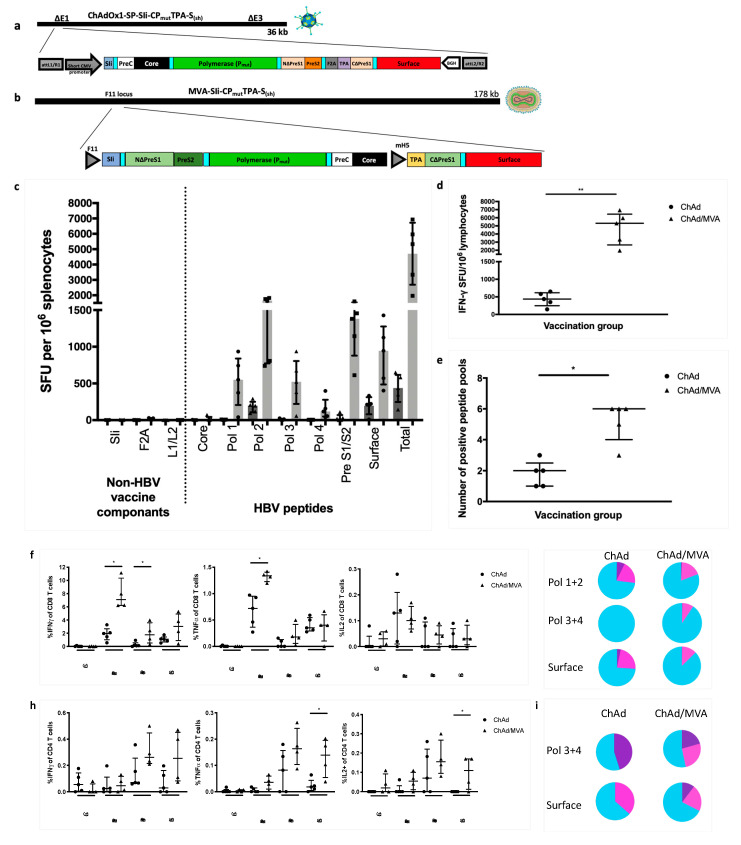
Priming with ChAdOx1-SP-SIi-CP_mut_TPA-S_(sh)_ and boosting with MVA-SIi-CP_mut_TPA-S_(sh)_ enhances the magnitude of vaccine induced T cell responses. C57BL/6J mice were vaccinated intramuscularly with 5 × 10^7^ IU of ChAdOx1-SP-SIi-CP_mut_TPA-S_(sh)_ alone (ChAd, *n* = 5, construct shown in (**a**)) or followed 7–8 weeks later by 2 × 10^6^ plaque forming units of MVA-SIi-CP_mut_TPA-S_(sh)_ (MVA, *n* = 5, construct shown in (**b**)). Splenocytes extracted 14 days after vaccination and plated in IFNγ ELISpot assays with pools of overlapping peptides corresponding to the vaccine immunogen (**c**). T cell response magnitude was measured by the number of IFNγ SFU per million lymphocytes (**d**) and breath was measured by the number of positive peptide pools (**e**) (over 100 IFNγ SFU/million cells defined as positive). Splenocytes extracted 14 days after vaccination were plated with pools of overlapping peptides corresponding to the vaccine immunogen and stained intracellularly for detection of cytokine production, shown as percentage of CD8+ cells producing IFNγ (left panel), TNFα (middle panel) or IL-2 (right panel) (**f**), proportion of CD8+ T cells that are single (blue), double (pink) or triple (purple) cytokine producers (**g**) percentage of CD4+ cells producing IFNγ (left panel), TNFα (middle panel) or IL-2 (right panel) (**h**) and proportion of CD4+ T cells that are single (blue), double (pink) or triple (purple) cytokine producers (**i**). Median and interquartile ranges are shown. Mann–Whitney tests were used for statistical comparison of medians between groups. * *p* < 0.05, ** *p* < 0.01.

**Figure 7 vaccines-08-00184-f007:**
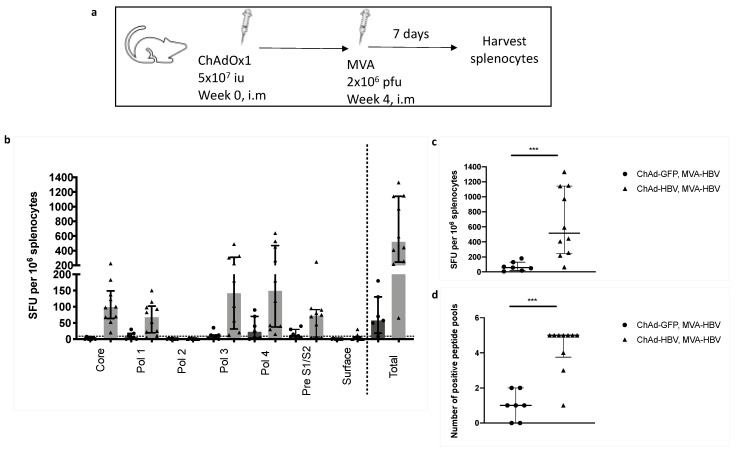
ChAd-MVA prime-boost vaccination in HHD mice induces T-cell response to HBV-core peptides. HHD mice were vaccinated intramuscularly (i.m) with 5 × 10^7^ iu (infectious units) of ChAdOx1-SP-SIi-CP_mut_TPA-S_(sh)_ or ChAdOx1-GFP as a negative control, followed 4 weeks later by 2 × 10^6^ pfu (plaque forming units) of MVA-SIi-CP_mut_TPA-S_(sh)_ (**a**)_._ T cell response magnitude was measured by the number of IFNγ SFU per million lymphocytes 7 days after MVA vaccination and plated in IFNγ ELISpot assays with pools of overlapping peptide corresponding to the vaccine immunogen (**b** and **c**) and breath was measured by the number of positive peptide pools (**d**) (over 20 IFNγ SFU/million cells defined as positive). Median and interquartile ranges are shown. Mann–Whitney tests were used for statistical comparison of medians between groups. *** *p* < 0.001.

**Figure 8 vaccines-08-00184-f008:**
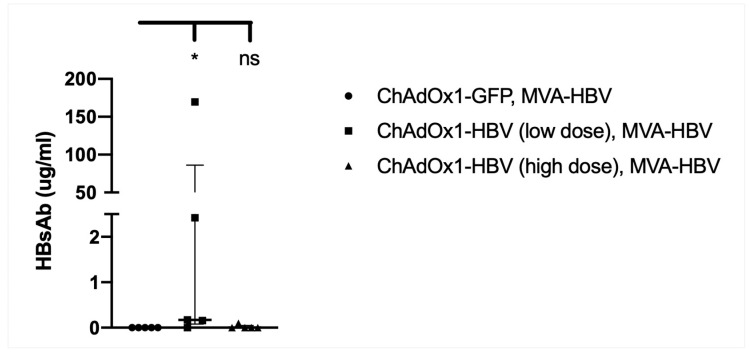
Vaccination with MVA-SIi-CP_mut_TPA-S_(sh)_ 7–8 weeks after ChAdOx1-SP-SIi-CP_mut_TPA-S_(sh)_ induces HBs-antibody. C57BL6 mice, *n* = 5 per group, were given intramuscularly injections with 5 × 10^5^ IU per mice (low dose) or 5 × 10^7^ IU per mice (high dose) of ChAdOx1-SP-SIi-CP_mut_-TPA-S_(sh)_ at week 0 followed by 2 × 10^6^ pfu per mice of MVA-SIi-HBV-CP_mut_TPA-S_sh_ at week 7–8. The third group of C57BL6 mice (*n* = 5), received 5 × 10^7^ IU per mice of ChAdOx1-GFP at week 0 followed by 2 × 10^6^ pfu per mice of MVA-HBV at week 8. Sera were collected 14 days post MVA vaccination and the levels of anti-HBs induction in response to vaccination was quantified by ELISA. Anti-HBs quantitation based on commercial standard mouse monoclonal antibodies to surface (GeneTex, GTX40707) above background value of naïve un-vaccinated sera are shown. Median and interquartile ranges are shown. The Kruskal–Wallis test and Dunn’s multiple comparisons test were used for statistical comparison of medians between groups. * *p* < 0.05, ns = not significant.

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
