# Peer review of "The Design and Development of a Multi-HBV Antigen Encoded in Chimpanzee Adenoviral and Modified Vaccinia Ankara Viral Vectors; A Novel Therapeutic Vaccine Strategy against HBV"

_vaccines, 2020, doi:10.3390/vaccines8020184_

Round 1
Reviewer 1 Report
This report by Chinnakannan et al demonstrates the design and development of HBV vaccine using Adenovirus and Vaccinia platforms. The results are interesting, well presented and important for the scientific community. My comments are outlined below-
- The abstract is too technical and not easy to read. Abstract should be a succinct summary of the work in layman terms (using minimal scientific jargons) that can inform a scientific as well as non-scientific readers. I would suggest modifying the abstract by summarizing the goal, results and importance of the study in a simple easy to read manner.
- On line 341, the authors mention that ChAdOx2 with long CMV promoter was unstable. Is there a particular region of the transgene that was getting deleted or was it random deletion?
- The use of CMV promoter typically results in very high expression of transgenes. It is good in terms of protein production but may also result in hyper immune response or toxicity in animal models. Can the authors comment on any negative impact on mice viability resulting from CMV driven transgene expression? Since HBV largely replicates in the liver, is it possible to use a liver specific promoter such as albumin to target expression of transgene only in liver cells?
- Have the authors tested various tissues like liver, kidney and heart for vaccine component expression?
- Can the authors comment on why BL/6J and BALB/C mice were used interchangeably during the study?
- Can the authors comment on Figure 8 as to why only low dose of the vaccine candidate is able to generate antibodies against the HBsAg?
- Minor comment- In Fig.6f and h, Ifn-gamma labeling (currently labeled INF) on Y-axis should be corrected. Fig7a, ChadOx1 dosage shown in im. Is this correct? Shouldn’t this be shown as PFU?
Author Response
Comments and Suggestions for Authors
This report by Chinnakannan et al demonstrates the design and development of HBV vaccine using Adenovirus and Vaccinia platforms. The results are interesting, well presented and important for the scientific community. My comments are outlined below-
- The abstract is too technical and not easy to read. Abstract should be a succinct summary of the work in layman terms (using minimal scientific jargons) that can inform a scientific as well as non-scientific readers. I would suggest modifying the abstract by summarizing the goal, results and importance of the study in a simple easy to read manner.
Thank you. We have now modified and simplified the abstract as suggested.
- On line 341, the authors mention that ChAdOx2 with long CMV promoter was unstable. Is there a particular region of the transgene that was getting deleted or was it random deletion?
To ensure generation of viral vectored HBV vaccines with an intact, stable, full-length HBV immunogen containing transgene cassette, we routinely perform quality control PCRs that amplifies the full-length transgene cassettes of all HBV vaccines. Only vaccines that show generation of a single PCR product, equivalent to the length of the HBV transgene cassette, were considered as stable HBV vaccines. Quality control PCR assay for ChAdOx2-SIi-CPmutS with long CMV promoter (ChAdOx2-LP-SIi-CPmutS) showed generation of multiple PCR products; in addition to a PCR product equivalent to the length of transgene cassette (5615 base pairs), four smaller PCR products (of approximately 500, 600, 900 and 1000 base pairs) were amplified from this vaccine. These smaller PCR products were considered as transgene deletions, which led us to conclude that ChAdOx2-LP-SIi-CPmutS as an unstable HBV vaccine. Since we managed to successfully generate 3 stable adenoviral HBV vaccines for this study (ChAdOx1-LP-SIi-CPmutS, ChAdOx1-SP-SIi-CPmutS and ChAdOx2-SP-SIi-CPmutS) we did not analyse the transgene deletions from ChAdOx2-SIi-CPmutS with long CMV promoter. We are unable to provide detailed information on whether it was a random or specific region deletion.
- The use of CMV promoter typically results in very high expression of transgenes. It is good in terms of protein production but may also result in hyper immune response or toxicity in animal models. Can the authors comment on any negative impact on mice viability resulting from CMV driven transgene expression? Since HBV largely replicates in the liver, is it possible to use a liver specific promoter such as albumin to target expression of transgene only in liver cells?
In all our mice immunogenicity studies that used CMV promoter driven HBV immunogens, we did not notice any negative impact on mice viability. For inducing a robust immune response to the vaccine immunogen, antigen presenting cells (APCs) are the major targets for the transgene (HBV immunogen) expression, rather than liver cells. So, we have used the CMV promoter, which is a ubiquitously active strong mammalian promoter. Also, immune response induction via intramuscular vaccination of viral vectored vaccines relies on immunogen expression in the APCs of the draining local lymphnodes. In this scenario, liver specific promoter is not an appropriate choice for transgene expression.
- Have the authors tested various tissues like liver, kidney and heart for vaccine component expression?
In addition to the work described in this manuscript, our group has a long standing history of working with human (adenovirus serotype 6 [Ad6]) and chimpanzee (adenovirus serotype 3 [ChAd3] and 63 [ChAd63]), adenovirus vectors, for the development of HCV vaccines [1-4]. In our earlier studies, we assessed tissue distribution of intramuscularly administered Ad6-HCV and showed localisation of this to the muscle and draining lymphnodes only, adjacent to the site of vaccine administration. Ad6 was not found to be present in other organs tested during the study (liver, spleen, ovary, teste and epididymis). This is not published, but is contained in data that we submitted to MHRA. Since then we have not repeated tissue distribution studies as the consensus/assumption now is that Ads given i.m are retained locally only.
Reference:
- Barnes, E.; Folgori, A.; Capone, S.; Swadling, L.; Aston, S.; Kurioka, A.; Meyer, J.; Huddart, R.; Smith, K.; Townsend, R., et al. Novel adenovirus-based vaccines induce broad and sustained T cell responses to HCV in man. Sci Transl Med 2012, 4, 115ra111, doi:10.1126/scitranslmed.3003155.
- Swadling, L.; Capone, S.; Antrobus, R.D.; Brown, A.; Richardson, R.; Newell, E.W.; Halliday, J.; Kelly, C.; Bowen, D.; Fergusson, J., et al. A human vaccine strategy based on chimpanzee adenoviral and MVA vectors that primes, boosts, and sustains functional HCV-specific T cell memory. Sci Transl Med 2014, 6, 261ra153, doi:10.1126/scitranslmed.3009185.
- Swadling, L.; Halliday, J.; Kelly, C.; Brown, A.; Capone, S.; Ansari, M.A.; Bonsall, D.; Richardson, R.; Hartnell, F.; Collier, J., et al. Highly-Immunogenic Virally-Vectored T-cell Vaccines Cannot Overcome Subversion of the T-cell Response by HCV during Chronic Infection. Vaccines (Basel) 2016, 4, doi:10.3390/vaccines4030027.
- Kelly, C.; Swadling, L.; Capone, S.; Brown, A.; Richardson, R.; Halliday, J.; von Delft, A.; Oo, Y.; Mutimer, D.; Kurioka, A., et al. Chronic hepatitis C viral infection subverts vaccine-induced T-cell immunity in humans. Hepatology 2016, 63, 1455-1470, doi:10.1002/hep.28294.
- Can the authors comment on why BL/6J and BALB/C mice were used interchangeably during the study?
We switched from BALBc mice (Figure 4) to C57BL/6 (Figure 6) as we are able to track CD8 restricted responses in C57BL/6 mice using a pentamer that identifies a particular epitope only found in the C57BL/6 mice. The pentamer data will form part of an in depth study of the immunological response, and the evaluation of tissue residency in a future study. This approach has allowed us to use a lower number of mice for the vaccine studies (in line with the reduce 3Rs principal of animal studies, https://www.nc3rs.org.uk/the-3rs) by being able to assess the immunological response in several ways concurrently.
- Can the authors comment on Figure 8 as to why only low dose of the vaccine candidate is able to generate antibodies against the HBsAg?
The reason why only the low dose of the vaccine candidate is able to generate antibodies against the HBsAg needs further investigation. One possibility of the generally poor antibody generation is that CD4 immunodominant epitopes in C57BL6 mice are not in the surface region of the HBV immunogen required to induce HBsAb. High dose prime with ChAdOx1 might preferentially induce responses towards immunodominant CD4 epitopes rather than subdominant epitopes. Conversely, low dose ChAdOx1 might prime some of the subdominant responses as well as the immunodominant responses, allowing a wider repertoire of CD4 epitopes to be boosted by MVA vaccination. If this were true, we might observe HBsAb responses in mice primes with low dose rather than high dose ChAdOx1. We have added a further sentence in the discussion to highlight this possibility (line 530 to 534).
Although we did detect anti-HBsAb in response to vaccination in some C57BL/6J mice, this was not ubiquitous, and titers varied. We show that the small S protein is secreted in vitro and that CD4+ surface-specific T cell responses are induced by vaccination. However, it is possible that CD4 immunodominant responses lie in regions other than the HBV surface region of the immunogen required for HBsAb generation and that antibodies are preferentially produced in response to the PreS1 region. Investigations of the surface-specific CD4+ T cell responses at the epitope level will be required to address this.
- Minor comment- In Fig.6f and h, Ifn-gamma labeling (currently labeled INF) on Y-axis should be corrected. Fig7a, ChadOx1 dosage shown in iu. Is this correct? Shouldn’t this be shown as PFU?
Thank you. Fig.6f and h, INF-gamma labelling on Y-axis has now been corrected to IFNg.
Since ChAdOx1 viral titers are based on iu, not PFU. ChAdOx1 dosage shown as iu (infectious units) in Fig7a is correct. Fig7a amended to make this clearer.
Figure 7 ledged amended as follows:
Figure 7. ChAd-MVA prime-boost vaccination in HHD mice induces T-cell response to HBV-core peptides. HHD mice were vaccinated intramuscularly (i.m) with 5x107 iu (infectious units) of ChAdOx1-SP-SIi-CPmutTPA-S(sh) or ChAdOx1-GFP as a negative control, followed 4 weeks later by 2x106 pfu (plaque forming units) of MVA-SIi-CPmutTPA-S(sh) (a).
Submission Date
02 March 2020
Date of this review
21 Mar 2020 06:23:09
Reviewer 2 Report
This study is well designed and well explained. The authors have done excellent job in presenting the idea and providing decent evidences. However, few more additions will help this manuscript to convey the message in a comprehensive manner.
- it would be helpful if authors can show the CD4 and CD8 T cells response that was specific for the vaccine.
- It has been shown that liver immune response may not be depicted by the Immune cells from blood or spleen. Hence a liver specific T cells response or IFN-g production would be add significant value to his study.
Author Response
Comments and Suggestions for Authors
This study is well designed and well explained. The authors have done excellent job in presenting the idea and providing decent evidences. However, few more additions will help this manuscript to convey the message in a comprehensive manner.
We would like to thank the reviewer for the positive comments.
- It would be helpful if authors can show the CD4 and CD8 T cells response that was specific for the vaccine.
CD4 and CD8 T response to the vaccine are shown in figures 6f and 6h, respectively. Descriptions are in line 477-490.
- It has been shown that liver immune response may not be depicted by the Immune cells from blood or spleen. Hence a liver specific T cells response or IFN-g production would be add significant value to his study.
Immune response to the vaccine measured from liver lymphocytes (intrahepatic lymphocytes) are shown in figures 3b and 3c. Descriptions are in line in 345-354.
Submission Date
02 March 2020
Date of this review
10 Mar 2020 04:57:51
Round 2
Reviewer 1 Report
The authors have satisfactorily addressed my concerns. I recommend acceptance of this manuscript for publication.